# Invasive Fungal Infections Complicating COVID-19: A Narrative Review

**DOI:** 10.3390/jof7110921

**Published:** 2021-10-29

**Authors:** Giacomo Casalini, Andrea Giacomelli, Annalisa Ridolfo, Cristina Gervasoni, Spinello Antinori

**Affiliations:** 1Luigi Sacco Department of Biomedical and Clinical Sciences, Università degli Studi di Milano, 20157 Milan, Italy; giacomo.casalini@unimi.it (G.C.); andrea.giacomelli@unimi.it (A.G.); 2III Division of Infectious Diseases, ASST Fatebenefratelli Sacco, 20157 Milan, Italy; annalisa.ridolfo@asst-fbf-sacco.it (A.R.); cristina.gervasoni@asst-fbf-sacco.it (C.G.)

**Keywords:** COVID-19, invasive fungal infections, aspergillosis, CAPA, candidemia, mucormycosis, CAM, PCP, histoplasmosis, cryptococcosis

## Abstract

Invasive fungal infections (IFIs) can complicate the clinical course of COVID-19 and are associated with a significant increase in mortality, especially in critically ill patients admitted to an intensive care unit (ICU). This narrative review concerns 4099 cases of IFIs in 58,784 COVID-19 patients involved in 168 studies. COVID-19-associated invasive pulmonary aspergillosis (CAPA) is a diagnostic challenge because its non-specific clinical/imaging features and the fact that the proposed clinically diagnostic algorithms do not really apply to COVID-19 patients. Forty-seven observational studies and 41 case reports have described a total of 478 CAPA cases that were mainly diagnosed on the basis of cultured respiratory specimens and/or biomarkers/molecular biology, usually without histopathological confirmation. Candidemia is a widely described secondary infection in critically ill patients undergoing prolonged hospitalisation, and the case reports and observational studies of 401 cases indicate high crude mortality rates of 56.1% and 74.8%, respectively. COVID-19 patients are often characterised by the presence of known risk factors for candidemia such as in-dwelling vascular catheters, mechanical ventilation, and broad-spectrum antibiotics. We also describe 3185 cases of mucormycosis (including 1549 cases of rhino-orbital mucormycosis (48.6%)), for which the main risk factor is a history of poorly controlled diabetes mellitus (>76%). Its diagnosis involves a histopathological examination of tissue biopsies, and its treatment requires anti-fungal therapy combined with aggressive surgical resection/debridement, but crude mortality rates are again high: 50.8% in case reports and 16% in observational studies. The presence of other secondary IFIs usually diagnosed in severely immunocompromised patients show that SARS-CoV-2 is capable of stunning the host immune system: 20 cases of *Pneumocystis jirovecii* pneumonia, 5 cases of cryptococcosis, 4 cases of histoplasmosis, 1 case of coccidioides infection, 1 case of pulmonary infection due to *Fusarium* spp., and 1 case of pulmonary infection due to *Scedosporium.*

## 1. Introduction

SARS-CoV-2, the causative agent of coronavirus disease 2019 (COVID-19), is responsible for a respiratory disease whose broad spectrum of severity ranges from asymptomatic or mildly symptomatic infection to severe bilateral pneumonia leading to progressive respiratory failure requiring non-invasive or invasive mechanical ventilation [1]. The death toll of COVID-19 is mainly due to extensive lung damage, although a number of studies have suggested that secondary pulmonary or non-pulmonary, bacterial, or fungal infections play a significant role [2,3]. Many factors may be associated with these secondary infections, including immune system dysregulation and inhibition, epithelial barrier damage, the widespread use of antibiotics, admission to an intensive care unit (ICU), mechanical ventilation, and prolonged hospitalisation [1,2,4]. The main etiological agents of secondary infections in COVID-19 are Gram-positive and Gram-negative bacteria, but some yeasts and moulds have also been described as deadly secondary pathogens [5].

Invasive pulmonary aspergillosis (IPA) is a life-threatening disease caused by the ubiquitous *Aspergillus* mould, which is typically associated with immune system dysregulation due to chronic glucocorticoid therapy, prolonged neutropenia in hematological patients, stem cell/solid organ transplantation, or immunosuppressive treatment [6]. Since the first reports of COVID-19 in Wuhan, some authors have highlighted the risk of invasive fungal infections (IFIs) in critically ill COVID-19 patients on the basis of the increasing number of reports of IPA in ICU patients without immunological disorders, such as those with severe influenza or chronic obstructive pulmonary disease [7,8,9]. Various published case reports and observational studies of COVID-19-associated IPA (CAPA) have raised concerns about the burden of this infection and difficulties in diagnosing it (non-specific clinical and radiological findings, the uncertain diagnostic performance of microbiological assays in an ICU, difficulties in differentiating colonisation from infections, the fact that the current clinical diagnostic algorithm does not really apply to COVID-19 patients, and concerns about collecting lower respiratory tract samples because of the risk of contagion) [10,11].

Candidemia is a typical secondary infection of critically ill patients that is associated with prolonged mechanical ventilation, total parenteral nutrition, broad spectrum antibiotic treatment, in-dwelling blood catheters, glucocorticoid therapy, and abdominal surgery before ICU admission [12,13]. COVID-19 patients admitted to an ICU generally undergo prolonged mechanical ventilation due to severe acute respiratory distress syndrome (ARDS) and receive antimicrobials, and are therefore at greater risk of developing secondary candidemia, which has been reported in a number of studies of COVID-19 [14,15].

Mucormycosis is an invasive fungal infection (IFI) that is caused by caused by naturally ubiquitous filamentous *Mucorales* fungi. The factors predisposing for mucormycosis include uncontrolled diabetes mellitus, glucocorticoid treatment, malnutrition, and other immunosuppressive conditions [16,17], and an increasing number of reports (mainly from India) have indicated that mucormycosis can be detrimental in patients with ongoing or previous COVID-19, especially if they are diabetic or have poorly controlled glycemia [18].

Finally, *Pneumocystis jirovecii* pneumonia (PCP), cryptococcosis, and histoplasmosis are other IFIs that have been reported in COVID-19 patients, which suggests that SARS-CoV-2-induced immune dysregulation can increase the likelihood of developing opportunistic infections typically seen in patients with severe immunodepression, such as acquired immunodeficiency syndrome (AIDS) or haematological malignancies [19].

The aim of this narrative review is to describe the characteristics of the IFIs complicating COVID-19 by separately analysing case reports and observational studies, with particular emphasis on the clinical and microbiological characteristics of CAPA and the clinical algorithms used to diagnose it.

## 2. Materials and Methods

The PubMed and Scopus databases were searched to find English, Italian, or Spanish case reports, case series, or retrospective or prospective observational studies of IFIs occurring in COVID-19 patients that were published between 1 January 2020 and 18 June 2021 using the following terms: (*Asp*ergill* OR CAPA) AND (invasive OR putative OR probable OR infection OR case OR patient OR report) AND (COVID* OR corona* OR SARS-CoV-2); (Candida OR candidemia OR candidiasis) AND (infection OR case OR patient OR report) AND (COVID* OR corona* OR SARS-CoV-2); (mucormyc* OR rhizopus) AND (invasive OR putative OR probable OR infection OR case OR patient OR report) AND (COVID* OR corona* OR SARS-CoV-2); (pneumocyst*) AND (infection OR case OR patient OR report) AND (COVID* OR corona* OR SARS-CoV-2); (cryptococc*) AND (infection OR case OR patient OR report) AND (COVID* OR corona* OR SARS-CoV-2); (histoplasm*) AND (infection OR case OR patient OR report) AND (COVID* OR corona* OR SARS-CoV-2). Adults diagnosed with IFIs and COVID-19 were included, regardless of their age or origin, and, when available, their patient-level demographic, clinical, and microbiological data were extracted. The different IFIs are discussed separately, differentiating case reports and case series from retrospective and prospective observational studies. We also discuss the characteristics of CAPA and the applicability of its various clinically diagnostic algorithms.

## 3. Results

### 3.1. COVID-19-Associated Pulmonary Aspergillosis

The literature search identified 88 articles describing cases of CAPA: 47 observational studies (28 retrospective and 19 prospective) [2,10,11,20,21,22,23,24,25,26,27,28,29,30,31,32,33,34,35,36,37,38,39,40,41,42,43,44,45,46,47,48,49,50,51,52,53,54,55,56,57,58,59,60,61,62,63] and 41 case report/series [64,65,66,67,68,69,70,71,72,73,74,75,76,77,78,79,80,81,82,83,84,85,86,87,88,89,90,91,92,93,94,95,96,97,98,99,100,101,102]. Fifty-three studies were from European countries (mainly from France, 16 articles (30.2%) and Italy, nine articles (17%)) and 34 from non-European countries (mainly the United States, six studies (17.6%)).

#### 3.1.1. Case Reports and Case Series

Seventy-four patients with CAPA were reported, of which 42 (56.8%) were classified as having “probable CAPA”, 12 (16.2%) as having “putative CAPA”, five (6.8%) as having “proven CAPA”, and 15 (20.3%) as having non-specified CAPA (Table 1; detailed information about the individual studies is given in Appendix A). Most of the studies were conducted in ICUs (92.7%), and 69 of the patients were mechanically ventilated (93.2%). The main diagnostic criteria used were *Asp*ICU [103] (seven studies) and modified *Asp*ICU (eight studies) [104]. The patients were predominantly male (49, 66.2%) and had a median age of 69 years (inter-quartile range (IQR) 57–74). The reported major co-morbidities were arterial hypertension (37 patients, 50%), diabetes mellitus (26, 35.1%), chronic pulmonary disease (13, 17.5%), cardiovascular disease (9, 12.2%), and obesity (4, 5.4%). It is worth noting chronic glucocorticoid exposure before COVID-19 and a CAPA diagnosis were reported in only three cases (one patient on chronic steroids for polymyalgia rheumatica, one solid organ transplant patient, and one patient with chronic obstructive pulmonary disease on inhaled steroids) [67,70,82]. Thirty-seven patients (50%) received corticosteroids for COVID-19 pneumonia, and 10 (13.5%) received at least one dose of tocilizumab. The median number of days from ICU admission or intubation to the diagnosis of CAPA was 13 (IQR 6–21).

Of the 72 patients for whom culture data were available, 62 had a respiratory culture positive for *Aspergillus* (86.1%), mainly bronchoalveolar lavage (25, 40.3%) and bronchial aspirate specimens (20, 32.3%). The most frequently isolated *Aspergillus* species were *A. fumigatus* (43 cultures, 67.2%) and *A. niger* (8 cultures, 12.5%), and the most frequent diagnostic samples were bronchoalveolar lavage (BAL) samples (25, 40.3%). Direct examination of respiratory samples showing septate branching hyphae was reported in 11 cases (17.7% of positive cultures) [65,66,67,71,76,86,87,88,89,92,96,102]. Proven CAPA was reported in five studies: in three cases, the diagnosis was made post portem [69,72,95]; in the remaining two cases, the diagnosis was made using pulmonary tissue (biopsy in one case, lobectomy in the other) [89,92]. Serum galactomannan (GM) was positive (optical density index [ODI] > 0.5) in 25/47 cases (53.2%) with a median value of 1.7 (IQR 1.1–3.1) (ODI index available in 21 cases). Respiratory sample GM was positive (ODI > 1) in 27/29 cases (93%), with a median value of 4.4 (IQR 2.2–6.1) (ODI index available in 23 cases); the median ODI of the BAL samples was 4.6 (IQR 2.2–6.3). Chest computed tomography (CT) findings suggesting invasive pulmonary aspergillosis (IPA) were reported in only 24/72 cases (33.3%).

Ninety-two percent of the patients received anti-fungal treatment, and there were two pan-azole-resistant infections [75,76]. In the case reported by Mohammed et al. [75], the resistance of the *A. fumigatus* strain was determined by means of phenotype testing using a 4-well triazole resistance screen (VIP check™) and the minimum inhibitory concentrations (MIC) using gradient strips. Ghelfenstein-Ferreira et al. described an *A. fumigatus* strain with high MICs for all azoles and identified the TR34/L98H mutation [76].

The overall reported mortality rate was 52.7% (39/64).

#### 3.1.2. Observational Studies

Four hundred and four cases of CAPA were described in 47 observational studies of 12,080 COVID-19 patients (average incidence 3.3%); the incidence of CAPA in the individual studies varied from 0.1% [28] to 57.1% [34] (Table 2; detailed information about the individual studies is given in Appendix A). Most of the studies (85%) exclusively involved critically ill patients requiring ICU admission.

Given the lack of consensus concerning the criteria for defining CAPA, it was diagnosed using various case definitions and clinical algorithms, including the *Asp*ICU algorithm with or without modifications (14 studies) [103], the modified *Asp*ICU criteria (7 studies) [104], the influenza-associated IPA (IAPA) criteria of Schauwvlieghe et al. (8 studies) [105], the new CAPA criteria of White et al. (1 study) [30], the algorithm proposed by ECCM/ISHAM (6 studies) [106], and the European Organization for Research and Treatment of Cancer and the Mycoses Study Group Education and Research Consortium (EORTC/MSGE) algorithm (7 studies) [107]. Ten studies did not specify the criteria used to diagnose CAPA [21,27,29,31,36,44,51,57,59,63].

The patients were mainly adult males (59.9%); had a median age ranging from 33 to 78 years; and were affected by arterial hypertension (40.1%), chronic cardio-pulmonary diseases (34.2%), and diabetes mellitus (27.9%). One hundred and thirty-nine (34.4%) received systemic corticosteroids, and 73 (18.1%) targeted COVID-19 treatment (anti-IL-6 drugs such as tocilizumab or eculizumab). Most of the patients were undergoing mechanical ventilation because of severe respiratory failure (84.7%).

The majority (58.3%) of the cultured samples were positive, with BAL samples being the most frequent (121/191 positive, 63.4%). The most frequently identified species was *Aspergillus fumigatus* (73.3%), followed by *A. niger* (5.3%), *A. flavus* (4.9%), and *A. terreus* (2%). Direct examination of respiratory sample showing septate branching hyphae was reported in 13 cases (4.9% of positive cultures), mainly BAL samples [24,33,35,43,47,52,54]. Serum GM was positive in only 70/379 cases (18.5%), with median ODI values ranging from 0.51 to 3.1. GM was positive in 157/272 respiratory samples (57.7%). One hundred and thirty-seven of the 239 tested BAL samples were GM positive (57.3%), with median ODI values ranging from 1 to 6.4. *Aspergillus*-specific polymerase chain reaction (PCR) testing of serum and respiratory samples was used in 16 studies and proved to be positive in 75.2% of the respiratory samples (mainly BAL, 69.5%) and 26% of the serum samples. Serum (1,3)β-D-glucan (BDG) was positive more frequently than serum GM (35%) (Table 2).

Histopathological or cytological findings were considered in eight studies involving 16 patients: fungal hyphae were detected in the respiratory specimens of 10 (nine BAL, one TA) [33,35,43,47,52,55], trans-bronchial pulmonary biopsy led to a diagnosis of “proven CAPA” in four [24], and the diagnosis was confirmed by means of autopsy in two [20].

As in the case reports and CAPA patient series, chest CT findings suggested IPA was rarely used in the observational studies (18.8% of cases): the main findings were nodules (69%), cavities (38.1%), and wedge-shaped consolidations (16.7%).

Anti-fungal treatment was received by 71.7% of the patients (voriconazole in 64.3% of cases). It is worth noting that two studies reported three cases of azole-resistant *Aspergillus*—Machado et al. described a voriconazole-resistant isolate of *A. lentulus* (MIC 2 mg/L) [39], and Meijer et al. described two azole-resistant isolates of *A. fumigatus* that were positive to the VIPCheck test and molecular biology for the TR34/L98H mutation [49].

The reported crude mortality rate in the observational studies was 54.6% (184/337).

#### 3.1.3. CAPA Classification Criteria

Three hundred and ninety-five CAPA cases were identified on the basis of the main classification criteria used by the authors of each study, and Appendix A summarises the classification criteria that we considered when we reclassified the CAPA cases using the patient-level data available in 35 studies (Appendix A) on the basis of the clinical algorithms shown in Table 3. The newly proposed ECMM/ISHAM definition criteria identified a higher number of CAPA cases (236 probable, 79 possible) [106]; the IAPA classification of Verweij et al. [104] identified 211 cases, and that of Schauwvlieghe et al. [105] identified 206 cases. The new CAPA criteria of White et al. [30] identified 180 putative cases of CAPA, whereas the *Asp*ICU algorithm [103] identified only 49 putative cases but 163 patients with *Aspergillus* colonisation (it is the only clinical algorithm incorporating a “colonisation” category). The diagnostic criteria proposed by the EORTC/MSGE [107], which were originally designed and validated in the context of haematological patients, identified only 25 cases (Table 4).

### 3.2. Candidemia and Other Yeast Fungemia

The literature search identified 41 articles reporting cases of yeast fungemia during COVID-19 [2,14,15,21,29,30,36,44,45,51,54,60,108,109,110,111,112,113,114,115,116,117,118,119,120,121,122,123,124,125,126,127,128,129,130,131,132,133,134,135,136]; most of them were from Italy (9, 22%), followed by the USA (6, 14.6%) and Brazil (4, 9.8%).

#### 3.2.1. Case Reports and Case Series

Nine case reports and seven case series were retrieved [108,109,110,111,112,113,114,115,116,117,118,119,120,121,122,123], together reporting 41 cases of candidemia (Table 4; detailed information about the individual studies is given in Appendix A). The median time from ICU admission or intubation to the diagnosis of candidemia was 13.5 days (IQR 7.5–29.5), and more than 80% of the patients had an in-dwelling vascular catheter, received broad-spectrum antibiotics, and underwent mechanical ventilation before its onset. Eight patients received at least one dose of tocilizumab, seven were treated with baricitinib, and more than half received systemic corticosteroids for COVID-19 pneumonia.

In 18 episodes, *Candida albicans* was isolated (40.9%), and 24 were due to other *Candida* species. In three patients from two studies, two *Candida* species were reported [108,119]. It is also worth noting that two patients were infected by *Saccharomyces cerevisiae* following treatment with a probiotic mixture for diarrhoea [118]. There were 11 cases of candidemia due to *C. auris*: two strains in the study of Allaw et al. and three in the study of De Almeida et al. were resistant to fluconazole and amphotericin B [112,114]; all of the six strains reported by Villanueva were resistant to amphotericin B and three were resistant to fluconazole [119]. A secondary infectious focus was described in four patients (one had endophthalmitis, one retinitis, one endocarditis, and one had fungal costochondritis and spondylitis confirmed by bone biopsy histology and culture) [109,113,123]. Anti-fungal treatment was received by 87.8% of the patients, mainly echinocandins (86%).

Twenty-three (56.1%) of the patients died.

#### 3.2.2. Observational Studies

Twenty-five observational studies reported 360 cases of candidemia, with an average incidence of 3.8% [2,14,15,21,29,30,36,44,45,51,54,60,124,125,126,127,128,129,130,131,132,133,134,135,136] (Table 5; detailed information about the individual studies is given in Appendix A). More than 80% of the patients had classic risk factors (an in-dwelling vascular catheter and broad-spectrum antibiotics), but only 21.4% received total parenteral nutrition. *C. albicans* and *C. glabrata* accounted for more than a half of the cases, but it is worth noting that there were two reports of fungemia due to *Tric**h**osporon asahii* and one case of *Rhodotorula mucillaginosa* fungemia [133,135,137]. The crude mortality rate was very high (74.8%, 202/270). Six studies reported infections due to resistant *Candida* strains: Chowdahary et al. found that 30% of their *C. auris* isolates were multi-azole (fluconazole + voriconazole)-resistant [124], Posteraro et al. described one strain of *C. glabrata* resistant to echinocandins [127], Cataldo et al. found a strain of *C. parapsilosis* resistant to fluconazole [129], Magnasco et al. described four strains of *C. auris* resistant to azoles and amphotericin B [132], Kayaslaan reported 30% of azole-resistant isolates, and Arasthefar found that two isolates were resistant to fluconazole and echinocandins [15,133]. Anti-fungal treatment was administered in 190 out of 253 cases (75%), consisting mainly of echinocandins (82 patients, 43.2%).

### 3.3. Mucormycosis

The literature search identified 45 articles reporting cases of COVID-19-associated mucormycosis [18,45,61,84,88,138,139,140,141,142,143,144,145,146,147,148,149,150,151,152,153,154,155,156,157,158,159,160,161,162,163,164,165,166,167,168,169,170,171,172,173,174,175,176,177]. Most of the articles were from India (19, 42%) and the USA (9, 20%); eight articles were from European countries.

#### 3.3.1. Case Reports and Case Series

Twenty-four case reports and nine case series [84,88,138,139,140,141,142,143,144,145,146,147,148,149,150,151,152,153,154,155,156,157,158,159,160,161,162,163,164,165,166,167,168] reported 59 cases of mucormycosis in patients with a median age of 51 years (IQR 42–62), 71.2% of whom were males (Table 6; detailed information about the individual studies is given in Appendix A). The most frequent co-morbidity was diabetes mellitus (77.9%).

The majority of cases (84.7%) were diagnosed during the clinical course of COVID-19, for which more half of the patients received systemic steroid treatment, and only 13.6% at least one dose of tocilizumab. The most frequent clinical presentations of mucormycosis (i9 cases each, 32.3%) were rhino-orbital [144,145,151,154,159,161,162,163,164,166] and rhino-orbito-cerebral [143,146,147,148,151,152,153,155,157,163,165], followed by 12 cases of pulmonary disease (20.3%) [84,88,138,139,141,144,147,149,167]. Autopsies of two cases revealed disseminated mucormycosis [144,158].

Histopathological examinations (HPEs) were carried out in all cases, 91.5% of which were positive; the diagnosis of the remaining five cases was supported by cultures of infected specimens (BAL, sputum, and purulent orbital material) [147,165]. Twenty-nine diagnoses of fungal rhinosinusitis were made on the basis of HPEs of nasal biopsies obtained during functional endoscopic sinus surgery (FESS); a further 10 diagnoses were based on HPEs of nasal material obtained by other means (discharges, scrapings, aspirates, or swabs). Definite diagnoses of pleuro-pulmonary mucormycosis were made after HPEs of BAL (three cases), a pulmonary biopsy (one case), pleural effusion (one case), a pleural biopsy (one case), induced sputum (one case), and BAL plus a pulmonary biopsy (one case) [84,88,138,139,141,145,147,149,167].

Cultures were positive in 33/37 patients (89.2%). The most frequent finding was *Rhizopus* spp. (26 isolates), with the strain being identified in 13 cases: *R. microsporus* (five cases), *R. arrhizus* (seven cases), and *R. azygosporus* (three cases). The species was identified using molecular biology techniques on tissue biopsies (positive PCR) in two cases [138,144]. Anti-fungal treatment was administered to 93.1% of the patients, and 59.3% required surgical interventions (31 maxillo-facial debridements, two thoracic soft tissue debridements, one pleural decortication, and one pulmonary resection).

Twenty-eight deaths were reported (50.8%).

#### 3.3.2. Observational Studies

The literature search found 12 observational studies, in which 3126 cases of mucormycosis were described [18,45,61,169,170,171,172,173,174,175,176,177], the average incidence of which was 8.6%. As shown in Table 7 (detailed information about individual studies is given in Appendix A), most of the patients were males (70.6%), had diabetes mellitus (76.1%), received systemic steroids for COVID-19 (73.8%), and developed mucormycosis concomitantly with COVID-19. In the largest Indian study, the median time from the diagnosis of COVID-19 to the diagnosis of mucormycosis was 13 days [177]. Rhino-orbital and rhino-sinusal mucormycosis were the most frequent (respectively, 48.9% and 24.6%); pulmonary involvement was reported in only 20 cases (0.6%). The diagnosis was confirmed by means of histopathology in 1340/3062 cases (43.8%), whereas cultures revealed zygomycetes in only 23.6%. The species was identified in 29 cases (5%) and not reported 508 cases. In one case, the diagnosis was made on the basis of a positive *Rhizopus*/*Mucor* PCR in a respiratory sample and serum [45]. More than two-thirds of the patients received anti-fungal treatment (75.6%), mainly amphotericin B (97.3%). A surgical intervention was undertaken in 2352/3113 patients (68.3%). All but two of the studies reported mortality data: the overall crude mortality rate was 16%.

### 3.4. Other Invasive Fungal Infections in COVID-19 Patients

Twenty-five studies (20 case reports/case series and five observational studies) describe uncommon IFIs in COVID-19 patients [30,45,61,110,137,148,178,179,180,181,182,183,184,185,186,187,188,189,190,191,192,193,194,195,196] (Table 8). Most are from non-European countries (seven from South America, 28%; six from the USA, 24%; single reports from Canada, China, and Qatar).

Twenty cases of *Pneumocystis jirovecii* pneumonia (PCP) were reported, all of them diagnosed during the course of COVID-19; 30% of the patients were HIV-positive at the time of both diagnoses. More than half of the patients required mechanical ventilation. The diagnosis of PCP was established mainly on the basis of *P. jirovecii* PCR in respiratory samples (80%); histopathological evidence was rare (one autoptic diagnosis, one direct examination of sputum) [183,187]. More than half of the patients received specific treatment (65%), and eight deaths were reported [178,179,180,181,182,183,184,185,186,187,188].

Finally, there were five reports of cryptococcal infections (four cases of disseminated cryptococcosis, one case of cryptococcal meningitis) [110,137,189,190,191], four cases of histoplasmosis (three disseminated, one pulmonary) [192,193,194,195], one case of pulmonary infection due to *Fusarium proliferatum* [45], one case of COVID-19 and coccidioides co-infection [196], and one pulmonary *Scedosporium* infection confirmed by a positive BAL culture [61].

It is worth noting that 10 patients had a history of immunosuppression (one with chronic hemolytic anemia, one who had undergone a solid organ transplantation, and eight with AIDS) [178,179,180,181,186,188,189,190,192,194,195].

## 4. Discussion

As expected, a number of cases of secondary bacterial and fungal infections have been reported since the emergence of COVID-19 and the publication of the first observational studies [90,130,197]; however, to the best of our knowledge, this is the only descriptive review of all of the types of IFI associated with COVID-19, the fungi involved, their clinical presentation and diagnostic difficulties, and their crude mortality rates. We identified 4099 cases of IFIs in 58,784 COVID-19 patients reported in 168 studies. CAPA and yeast fungemia were mainly reported among critically ill COVID-19 patients requiring ICU admission (in respectively 88% and 65% of the studies), whereas only 47% of the mucormycosis studies were carried out in an ICU.

Although it is currently unknown whether SARS-CoV-2 is an independent risk factor, patients with severe COVID-19 seem to be at risk of developing IPA. We identified a total of 478 cases of CAPA in 47 observational studies and 41 case reports/case series. Its average incidence was 3.9%, but its incidence in the observational studies ranged from 0.1% [28] to 57% [34], a variability that is probably due to differences in the diagnostic strategies and clinical algorithms used by the authors.

It is difficult to diagnose IPA in ICU patients as all of the proposed diagnostic algorithms have major limitations (Appendix A). The EORTC/MSG criteria do not really apply to COVID-19 patients because they often do not have any host risk factors and the CT criteria are infrequently observed in mechanically ventilated patients [107]. Among the patients we analysed, the EORTC/MSG criteria identified only 25 cases of CAPA (16 probable and 9 possible: 6.3%). In a bid to overcome such limitations, Blot et al. validated the *Asp*ICU algorithm, which distinguishes putative IPA from *Aspergillus* colonisation (the other algorithms do not contemplate a “colonisation” category); however, although its diagnostic performance is better than that of the EORTC/MSG criteria, its main limitation is that it does not include indirect biomarkers of fungal infection (GM or BDG) [103], and it identified only 49 putative cases of CAPA (12.4%).

The two case definitions of influenza-associated pulmonary aspergillosis (IAPA) in ICU patients proposed by Verweij and Schauwvlieghe allowed for a probable CAPA diagnosis in COVID-19 patients admitted to an ICU with pulmonary infiltrates and a positive BAL culture or BAL or serum GM [104,105]. These criteria are less stringent than the EORTC/MSG and *Asp*ICU criteria, but they may lead to an over-estimate of the incidence of CAPA and are less specific as patients with just a single positive GM test and chest X-ray infiltrates (frequent findings in critically ill COVID-19 patients) may be classified as having CAPA: applying these two criteria to all of the CAPA studies led to the identification of, respectively, 211 (53.4%) and 206 cases (52.2%), the second highest numbers of all of the classification criteria. Moreover, neither case definition includes the PCR testing of serum or respiratory sample or BDG, whereas it has been shown (at least in hematological patients) that the diagnostic performance of combining multiple biomarkers is better than using just one [198].

White et al. have proposed a new CAPA definition based on radiological and mycological criteria, which includes *Aspergillus* PCR testing and serum BDG, which in one study of ICU patients has shown to be more sensitive than serum GM in diagnosing aspergillosis [199]. However, although one biomarker may be diagnostically sufficient in the presence of a specific radiological feature, this new CAPA definition requires at least two positive mycology results to be sufficiently specific in the presence of non-specific chest radiology findings [30]. Nevertheless, in comparison with *Asp*ICU, it identified 131 more cases. 

Finally, the consensus definition proposed by ECMM/ISHAM, which introduced the use of a lateral flow device or assay (LFD/LFA) as a mycological criterion and serial PCR testing of serum with the aim of increasing specificity, identified 315 CAPA cases (236 probable and 79 possible), the highest number of all of the classifications [106]. However, this was probably because it also includes other kinds of respiratory samples such as BAS or TA that inevitably decrease specificity despite their sensitivity, thus raising doubts about its ability to distinguish real cases of IPA from *Aspergillus* colonisation. For example, using the ECMM/ISHAM criteria, some of the CAPA cases reported by Alanio, Fekkar, and Rabagliati would be classified as probable CAPA on the basis of only one or two mycological criteria, whereas the authors state that these patients survived without any anti-fungal treatment, thus questioning the validity of the CAPA diagnosis [11,45,61], something that is also confirmed by applying the Verweij and Schauwvlieghe algorithms to the same patients.

The real incidence of CAPA is therefore difficult to estimate, regardless of which clinical algorithm is used. A systematic review by Chong et al., which only considered observational studies using well-defined diagnostic criteria, found an overall incidence of 13.5% (range 2.5–35%) among COVID-19 patients [200], whereas Mitaka et al. found an incidence of 10.2% in ICU patients [201]. CAPA rates probably depend on patient selection and the study setting (mainly ICUs in the studies analysed here), and therefore it is difficult to make meaningful comparisons. For example, studies of ICU patients may suffer from “sample bias” (easier access to lower respiratory tract specimens, enhanced testing in screening programmes for fungal diseases), whereas, in other settings, the limited availability of diagnostic marker testing (PCR testing and serum BDG) and limited awareness of the disease may lead to a significant under-estimate.

Only 62% of the CAPA cases had a positive respiratory culture, and the specimen was BAL in 45% of cases, which means that a large proportion of the diagnoses were based on cultures of less informative respiratory specimens. Only a few studies provided data concerning direct microscopic examinations of respiratory samples, and a positive result (septate branching hyphae suggesting *Aspergillus*) was reported in only 24 cases (7.4% of the positive cultures: 16 BAL, 5 TA, 1 BAS, and 2 unspecified respiratory specimens) [24,33,35,43,47,52,54,65,66,67,71,76,86,87,88,89,92,96,102]. According to Blot et al., positive direct examination of BAL is a crucial criterion for a diagnosis of putative IPA in ICU patients without host risk factors [103]. Given the paucity of direct microscopy data, the diagnostic yield of Blot’s algorithm is low, although some authors considered COVID-19 a host risk factor and thus expanded its diagnostic capacity [22].

The diagnostic performance of the *Aspergillus*-specific PCR testing of respiratory samples was better than that of standard cultures (78.3% considering case reports, case series, and observational studies together). Molecular biology techniques may provide added value when diagnosing IPA, even though most of the data come from adults with haematological malignancies [107], and usefulness of PCR testing in COVID-19 patients has not yet been demonstrated.

The diagnostic performance of serum GM (which ranged from 18% in the observational studies to 53.1% in the case reports; 22.4% overall) was much lower than that of BAL GM (60%), thus confirming the poor sensitivity of the assay in non-neutropenic patients [202]. Moreover, a study by Farmakiotis et al. has shown that BAL GM has a high false positivity rate in patients without haematological malignancies [203]. In 10 studies, GM was tested in respiratory samples other than BAL (BAS, TA, or sputum) and had a positivity ratio of 61.5%; however, in these specimens, GM and PCR testing have not been fully validated, sample collection is not targeted, and it is difficult to establish whether a positive culture corresponds to colonisation or infection and tissue invasion [25,41,49,55]. Serum BDG is apparently diagnostically better than serum GM and has a sensitivity of 36.7% (84/229), but this pan-fungal assay is not specific for aspergillosis.

Making a radiological diagnosis of IPA in COVID-19 patients is challenging for a number of reasons: (1) there numerous technical difficulties in obtaining CT images of patients admitted to an ICU, (2) the radiological features of COVID-19 can mask IPA, and (3) the low frequency of imaging suggesting IPA in mechanically ventilated patients [204]. As only 18.8% of the patients in observational studies and 33.3% of those described in case reports showed imaging features suggesting IPA, it seems that radiological criteria are of little help in diagnosing CAPA.

Finally, it must be remembered that only a few studies attempted to confirm the clinical diagnosis of CAPA by means of HPEs. Only 21 of our reported CAPA cases were histopathologically confirmed, mainly because invasive sampling is frequently unfeasible in COVID-19 patients, especially those who are if critically ill and admitted to an ICU. A recent systematic review of autopsy studies found that the prevalence of autopsy-proven invasive mould disease was only 2% (11/677 deaths, of which 8 were diagnosed as having CAPA [205]), a much lower value than the CAPA rates reported in some observational studies of ICU patients [11,24,30]. As HPE is the reference standard for diagnosing IFIs, the CAPA incidence rates in observational studies are probably over-estimated and the specificity of clinical algorithms is low. On the other hand, autopsies carried out at a single centre in Germany showed four out of eight patients in whom an IFIs was not clinically suspected actually had autoptic signs of an IFI that was considered the cause of death: two cases of CAPA and two of mucormycosis [206].

In brief, the real burden of CAPA has not yet been established as the currently available clinical tools are not sensitive or specific enough to detect it.

The average incidence of candidemia in the observational studies was 3.8%, and the time between ICU admission and a diagnosis was 13.5 days. The latter is similar to the pooled mean duration of ICU admission before the onset of candidemia found by a recent meta-analysis (12.9 days) [207]. Nearly 30% of the case report/case series came from the Middle East, whereas the highest number of observational studies of candidemia came from Italy (33%) and the USA (16.7%). A very high average crude mortality rate was observed in the observational studies (74.8%), but it is worth noting that the ICU outcomes candidemia patients were worse than those observed CAPA patients even before the COVID-19 pandemic [208]. As recently pointed out by McCarty, it is difficult to consider COVID-19 patients hospitalised in an ICU and diagnosed with candidemia as a “truly unique patient population” because most of the patients had classic risk factors (i.e., an in-dwelling vascular catheter, antibiotic treatment, and mechanical ventilation) [209]. However, as 43% received steroids, 19.1% received tocilizumab, and 25.3% were affected by diabetes, it is possible that drug-induced or a metabolic-associated condition was responsible for an increased risk of infection.

The exact global burden of mucormycosis is not known because there is a lack of population-based studies, but its incidence is increasing, with India accounting for most of the cases worldwide [210,211]. It is worth noting that an unusually high number of mucormycosis cases in COVID-19 patients was reported during the second surge of the pandemic in India. Our search identified a total of 3185 cases distributed in 33 case reports/series and 12 observational studies, with an average incidence of 8.7%; many of these came from India (12/33 case reports, 36%, and 7/12 observational studies, 58.3%). Rhino-orbital mucormycosis was the most frequently reported (48.6% overall), and the strong association between COVID-19 and mucormycosis in the Indian population is probably due to a convergence of multiple concomitant risk factors: a higher baseline prevalence of the disease, favourable environmental factors (humidity, soil contamination, air pollution), a high and increasing prevalence of uncontrolled diabetes mellitus, and the widespread use of corticosteroids to treat COVID-19 (often beyond the indications of the guidelines) and possibly other immunosuppressive agents (i.e., anti-IL-6 agents) [176,212]. More than 76% of the candidemia patients considered in this review were affected by diabetes mellitus with poor glycaemic control, and almost 73% received systemic corticosteroids, which may worsen glycaemic control and presumably reduce host immune defences. The characteristics of the candidemia cases (disease localisation, etiologic agents, diagnostic methods, treatment, and outcomes) were similar to those with other IFI [211]. The observational studies indicate a low rate of histopathological diagnoses (43.8%) and positive cultures (23.6%), even though these are required for a proven diagnosis [213]. This is probably due to technical difficulties in obtaining diagnostic samples from COVID-19 patients because of safety concerns or clinical instability secondary to respiratory failure. Furthermore, *Mucorales* are fragile and, if the material taken from biopsies or other specimen is not managed carefully, a culture may become negative [213]. The treatment of mucormycosis is challenging and requires the prompt administration of anti-fungal agents and debulking surgery but, even after treatment, the average mortality rate associated with the disease is 40% [213]. Among the case reports/case series, maxillofacial surgery was reported in 35 cases (59.3%); however, the outcome was not always favourable probably because of highly invasive disease [140,145,147,148,149,150,151,157,159,161,163]. More than one-third of the patients in the observational studies underwent surgical debridement. In the largest study of COVID-19-associated mucormycosis, a disease stage of <3b (orbital involvement without vision loss) was associated with a better outcome and, regardless of stage, surgery (paranasal sinus debridement and/or orbital exenteration) was associated with stable residual or regressive disease [177]. Together with infection control, the key elements of successful mucormycosis management suggested in the clinical guidelines are an early diagnosis, species identification, a combination of anti-fungal therapy and aggressive surgical resection/debridement, the optimisation of blood glucose levels, and the correct use of glucocorticoids for COVID-19 treatment (administration to hypoxic patients, and not exceeding the dose and treatment duration established by the guidelines) [213,214,215,216].

A few reports of other IFIs complicating COVID-19 (PCP, cryptococcosis, and histoplasmosis) have been published, but their burden is much less than that of CAPA, candidemia, or mucormycosis. However, physicians should be aware of the immunosuppressive potential of SARS-CoV-2 and consider these diagnoses in patients with suggestive signs and symptoms who experience an unexplained clinical worsening in COVID-19, even in the absence of severe baseline immunodepression [1,19,182]. Diagnosing PCP in COVID-19 patients is difficult and requires a high degree of clinical suspicion as they share some clinical features (signs and symptoms, pulmonary ground glass opacities in CT images, high levels of lactate dehydrogenase (LDH)). Positive *Pneumocystis* PCR tests of oropharyngeal wash samples or sputum probably over-estimate the real burden of the disease, and therefore treatment with cotrimoxazole may not always be indicated [183].

## 5. Conclusions

Increasing published evidence should warn physicians about the association between COVID-19 and IFIs. An increasing number of case reports and observational studies have shown that the clinical course of COVID-19 can be complicated by a variety of fungal super-infections with unfavourable outcomes. Although CAPA, CAM, and candidemia have received the most attention, physicians should remember that the invasive mycoses typically observed in highly immunodepressed patients can also be found in COVID-19 patients. One of the outstanding questions remains the true burden of fungal infections in COVID-19 patients, but the answer can only come from prospective studies that make an extra effort to obtain histopathological confirmation in order to reach a definite diagnosis.

## Figures and Tables

**Table 1 jof-07-00921-t001:** Case reports and case series describing COVID-19-associated pulmonary aspergillosis (CAPA).

Characteristics	No. (%)
CAPA cases	74
CAPA definitions	Probable: 42 (56.8) Putative: 12 (16.2) Proven: 5 (6.8) Not reported: 15 (20.3)
Sex	Male: 49 (66.2)
Median age (IQR)	69 (57–74)
Main underlying diseases	Arterial hypertension: 37 (50) Diabetes mellitus: 26 (35.1) Chronic pulmonary disease: 13 (17.6) Cardiovascular disease: 9 (12.1) Obesity: 5 (6.8)
COVID-19 treatments	Steroids: 37 (50) Tocilizumab: 10 (13.5) Eculizumab: 2 (2.7) Siltuximab: 1 (1.4); anakinra: 1 (1.4)
Median number of days from ICU admission or TI to CAPA diagnosis (IQR)	13 (6–21)
No. of positive cultured samples/tested samples Sample types	62/72 (86.1) BAL: 25 (40.3); 6 with positive DE BAS: 19 (30.6); 1 with positive DE TA: 14 (22.6); 4 with positive DE Sputum: 2 (3.2); autopsy: 1 (1.6); biopsy: 1 (1.6)
*Aspergillus* species	*A. fumigatus*: 43 (67.2) *A. niger*: 8 (12.5) *A. flavus*: 4 (6.3) *A. terreus*: 3 (4.7) Others: 6 (9.4). Total: 64
No. of GM-positive samples/tested samples: median ODI (IQR)	Serum: 25/47 (53.2); ODI: 1.7 (1.1–3.1) Respiratory samples: 27/29 (93); ODI 4.4 (2.2–6.1) BAL: 4.6 (2.2–6.3) 21/21: BAL; 4/6: TA; 1/1: BAS; 1/1: sputum
No. of PCR-positive samples/tested samples	Respiratory: 8/10 (80) 3/3: BAL; 2/2: TA; 1/1: BAS Serum: 2/4 (50)
No. of BDG-positive samples/tested samples	10/19 (52.6)
No. of patients with specific imaging findings	24/72 (33.3) Nodules: 13 (54.2) Cavities: 10 (41.7) Air crescent: 2 (8.3) Halo: 2 (8.3) Balls: 1 (4.1); cysts: 1 (4.1) Tree-in-bud: 1 (4.1)
No. of patients receiving anti-fungal treatment Anti-fungal agents	68 (92) Voriconazole: 35 (51.5) AMB: 15 (22.1) Echinocandin: 15 (22.1) Isavuconazole: 6 (8.8)
No. of deaths	39 (52.7)

Data are absolute numbers (percentage) unless otherwise specified. Abbreviations: IQR, interquartile range; ICU, intensive care unit; TI, tracheal intubation; GM, galattomannan; ODI, optical density index; BAL, bronchial-alveolar lavage; BAS, bronchial aspirate; TA, tracheal aspirate; BDG, (1–3)-β-D-glucan; AMB, amphotericin B.

**Table 2 jof-07-00921-t002:** Observational studies reporting CAPA.

Characteristics	No. (%)
No. of CAPA cases/total number of patients	404/12,080 (3.3)
Sex	Male: 242 (59.9)
Main underlying diseases	Arterial hypertension: 162 (40.1) Diabetes mellitus: 113 (27.9) Chronic pulmonary disease: 84 (20.8) Cardiovascular disease: 54 (13.4) Obesity: 95 (23.5)
COVID-19 treatments	Steroids: 139 (34.4) Tocilizumab: 71 (17.6) Eculizumab: 2 (0.5)
No. of positive cultured samples/tested samples Sample types	261/448 (58.3) 121/191: BAL (10 with positive DE) 38/40: BAS 53/74: TA (1 with positive DE) 9/18: sputum 44/49: respiratory specimen NS (2 with positive DE)
*Aspergillus* species	*A. fumigatus*: 181 (73.3) *A. niger*: 13 (5.3) *A. flavus*: 12 (4.9) *A. terreus*: 5 (2) Others: 36 (14.6) Total: 247
No. of GM-positive samples/tested samples	Serum 70/379 (18.5) Respiratory samples 157/272 (57.7) 137/239: BAL 14/21: TA 4/10: BAS 2/2: respiratory specimen NS
No. of PCR-positive samples/tested samples	Respiratory samples 100/133 (75.2) 57/82: BAL; 10/14: TA; 16/18: BAS; 17/19: respiratory specimen NS Serum 13/50 (26)
No. of BDG-positive samples/tested samples	74/210 (35)
No. of patients with specific imaging findings	42/223 (18.8) Nodules: 29 (69) Cavities: 16 (38.1) Wedge-shaped consolidations: 7 (16.7) Air crescent: 2 (4.8) Halo sign: 2 (4.8)
No. of patients receiving anti-fungal treatment Anti-fungal agents	230/321 (71.7) Voriconazole: 148 (64.3) AMB: 55 (23.9) Echinocandin: 30 (13) Isavuconazole: 27 (11.7) Other azoles: 7 (3)
No. of deaths	184/337 (54.6)

Data are absolute numbers (percentage) unless otherwise specified. Positive direct examination refers to the observation of septate branching hyphae in respiratory samples. Abbreviations. GM, galattomannan; ODI, optical density index; BAL, bronchial-alveolar lavage; BAS, bronchial aspirate; TA, tracheal aspirate; NS, not specified; BDG, (1–3)-β-D-glucan; AMB, amphotericin B.

**Table 3 jof-07-00921-t003:** Comparison of the different CAPA diagnostic algorithms.

Author	Author’s Classification/No. of Cases	*Asp*ICU (Blot et al.)	Modified *Asp*ICU (Verweij et al.)	IAPA (Schauwvlieghe et al.)	ECMM/ISHAM (Koehler et al.)	CAPA (White et al.)	EORTC (Donnelly et al.)
Bartoletti et al.	mAspICU/ 30 probable	2 proven 28 NC	28 probable 2 proven	28 probable 2 proven	28 probable 2 proven	17 putative 2 proven 11 NC	2 proven 28 NC
Chen et al.	NR/ 1 CAPA NS	No data	No data	No data	No data	No data	No data
Koehler et al.	AspICU personal/ 5 putative	3 colonisation 2 NC	4 probable 1 NC	4 probable 1 NC	4 probable 1 possible	5 putative	5 NC
Lahmer et al.	AspICU personal/ 11 putative	11 colonisation	11 probable	11 probable	11 probable	9 putative 2 NC	11 NC
Rutsaert et al.	AspICU/ 4 proven 3 putative	4 proven 1 putative 1 colonisation 1 NC	4 proven 2 probable	4 proven 2 probable	4 proven 2 probable 1 possible	4 proven 1 putative 2 NC	4 proven 3 NC
Alanio et al.	AspICU personal/ 9 putative	2 putative 5 colonisation 2 NC	6 probable 3 NC	6 probable 3 NC	6 probable 2 possible 1 NC	6 putative 3 NC	1 probable 8 NC
Brown et al.	IAPA/ 2 probable	2 colonisation	2 NC	2 probable	2 possible	2 putative	2 NC
Van Arkel et al.	mAspICU/ 3 probable	4 colonisation 2 NC	3 probable 3 NC	3 probable 3 NC	3 probable 2 possible 1 NC	3 putative 3 NC	6 NC
Nasir et al.	NR/ 5 putative 4 colonisation	9 colonisation	9 NC	9 NC	5 possible	2 putative 3 NC	5 NC
Gangneux et al.	AspICU pers./ 4 putative 3 probable	9 putative 6 NC	7 probable 8 NC	7 probable 8 NC	15 probable	7 putative 8 NC	1 probable 14 NC
Falces-Romero et al.	AspICU/EORTC/ 7 putative 1 probable 2 NC	9 colonised 1 NC	2 probable 8 NC	2 probable 8 NC	2 probable 7 possible 1 NC	2 putative 8 NC	1 probable 9 NC
Zhang et al.	NR/ 3 CAPA NS	No data	No data	No data	No data	No data	No data
White et al.	CAPA/ 19 putative	8 putative 3 colonised 14 NC	20 probable 5 NC	20 probable 5 NC	22 probable 3 NC	19 putative	1 probable 24 NC
Ichai et al.	NR/ 6 probable/proven	No data	No data	No data	No data	No data	No data
Sarrazyn et al.	AspICU/ 4 putative	4 colonisation	4 probable	4 probable	4 probable	4 putative	4 NC
Dupont et al.	AspICU pers./ 19 putative	2 putative 14 colonised 3 NC	12 probable 7 NC	11 probable 8 NC	11 probable 8 possible	4 putative 15 NC	19 NC
Mitaka et al.	AspICU/ 4 putative	4 putative	4 probable	4 probable	4 probable	1 putative 3 NC	4 NC
Borman et al.	AspICU pers./ 14 probable 1 possible	1 putative 3 colonised 11 NC	8 probable 7 NC	8 probable 7 NC	9 probable 1 possible 5 NC	9 putative 6 NC	15 NC
Rothe et al.	NR/ 9 CAPA NS	No data	No data	No data	No data	No data	No data
Wang et al.	EORTC/ 8 possible	4 colonisation 4 NC	4 probable 4 NC	4 probable 4 NC	4 probable 4 NC	4 putative 4 NC	8 possible
Chauvet et al.	Multiple/ 5 putative 1 possible	3 putative 1 colonised 2 NC	3 probable 3 NC	3 probable 3 NC	3 probable 2 possible 1 NC	2 putative 4 NC	1 probable 1 possible 4 NC
Machado et al.	AspICU/ 8 putative	8 putative	6 probable 2 NC	6 probable 2 NC	6 probable 2 possible	6 putative 2 NC	1 probable 5 NC
Segrelles-Calvo et al.	EORTC/ 7 probable	No data	No data	No data	No data	No data	No data
Roman-Montes et al.	IAPA/ 14 probable	10 colonised 4 NC	6 probable 8 NC	6 probable 8 NC	6 probable 8 possible	13 putative 1 NC	14 NC
Permpalung et al.	Personal/ 20 probable 19 possible	No data	No data	No data	No data	No data	No data
Delliere et al.	mAspICU + EORTC/ 21 probable	2 putative 17 colonised 2 NC	19 probable 3 NC	16 probable 5 NC	16 probable 5 possible	13 putative 8 NC	2 probable 19 NC
Falcone et al.	NR/ 1 CAPA NS	No data	No data	No data	No data	No data	No data
Fekkar et al.	EORTC/ 6 probable	4 putative 2 colonised	4 probable 2 NC	4 probable 2 NC	5 probable 1 possible	6 putative	4 probable 2 NC
Gouzien et al.	mAspICU/ 2 probable	1 putative 1 colonised	2 probable	2 probable	2 probable	2 NC	2 probable
Van Grootveld et al.	ECMM/ 11 probable 8 NC	1 putative 10 colonised 8 NC	10 probable 9 NC	10 probable 9 NC	11 probable 4 possible 4 NC	10 putative 9 NC	19 NC
Maes et al.	IAPA/ 3 probable	3 NC	3 probable	3 probable	3 probable	3 NC	3 NC
Meijer et al.	ECMM/ 13 probable	13 colonised	8 probable 5 NC	8 probable 5 NC	8 probable 5 possible	10 putative 3 NC	13 NC
Van Biesen et al.	Personal/ 9 putative	1 putative 6 colonised 2 NC	9 probable	9 probable	9 probable	7 putative	9 NC
Obata et al.	NR/ 4 CAPA NS	No data	No data	No data	No data	No data	No data
Razazi et al.	mAspICU/IAPA 7 probable	2 putative 4 colonised 1 NC	6 probable 1 NC	6 probable 1 NC	6 probable 1 NC	7 NC	7 NC
Ripa et al.	AspICU personal/ 11 putative	No data	No data	No data	No data	No data	No data
Pintado et al.	ECMM/ 16 probable	No data	No data	No data	16 probable	No data	No data
Versych et al.	IAPA/ 2 probable	2 colonised	2 probable	2 probable	2 probable	2 putative	2 NC
Reizine et al.	ECMM/ 10 probable	9 colonised 1 NC	3 probable 7 NC	3 probable 7 NC	4 probable 6 possible	10 putative	10 NC
Oliva et al.	mAspICU/IAPA 2 probable	2 NC	2 probable	2 probable	1 probable 1 NC	2 NC	2 NC
Campochiaro et al.	NR/ 1 CAPA NS	1 colonised	1 probable	1 probable	1 probable	1 NC	1 NC
Wasylyshyn et al.	ECMM/ 3 probable	2 colonised 1 NC	2 probable 1 NC	2 probable 1 NC	2 probable 1 possible	2 putative 1 NC	2 NC
Nebreda-Moyoral et al.	NR/ 3 CAPA	3 colonised	3 NC	3 NC	3 possible	3 NC	3 NC
Signorini et al.	EORTC/ 2 probable	No data	No data	No data	No data	No data	2 probable
Rabagliati et al.	ECMM/ 14 probable	7 colonised 7 NC	8 probable 6 NC	8 probable 6 NC	7 probable 7 possible	4 putative 10 NC	14 NC
Lamoth et al.	mAspICU personal/ 1 probable 2 putative	3 colonised	1 probable 2 NC	1 probable 2 NC	1 probable 2 possible	2 putative	3 NC
Yang et al.	NR/ 2 CAPA NS	No data	No data	No data	No data	No data	No data
Total	395 cases, of which 6 proven	49 putative 163 colonised	211 probable	206 probable	236 probable 79 possible	180 putative	16 probable 9 possible

Abbreviations. NS, not specified; NC, not classifiable.

**Table 4 jof-07-00921-t004:** Case reports and case series describing candidemia associated with COVID-19.

Characteristics	No. (%)
Candidemia cases	41
Sex	Male: 32 (78)
Median age (IQR)	64 (54–72)
Main underlying diseases	Diabetes mellitus: 27 (65.9) Arterial hypertension: 24 (58.5) Cardiovascular disease: 9 (21.9) Obesity: 9 (21.9) Chronic pulmonary disease: 4 (9.8)
COVID-19 treatments	Steroids: 27 (65.9) Tocilizumab: 8 (19.7) Baricitinib: 7 (17.1)
Median number of days from ICU admission or TI to candidemia (IQR)	13.5 (7.5–29.5)
No. of patients with central venous catheters	35 (85.4)
No. of patients on total parenteral nutrition	5 (12.8)
No. of patients on antibiotic therapy	34 (82.9)
No. of patients undergoing mechanical ventilation	36 (87.8)
*Candida* species	*C. albicans:* 18 (40.9) *C. auris*: 11 (26.8) *C. glabrata:* 6 (14.6) *C. tropicalis:* 4 (9.8) Other: 5 (12.2) Total: 44
No. of patients with a secondary focus	Retinitis: 1 Endocarditis: 1 Endophathalmitis: 1 Osteomyelitis: 1
No. patients receiving anti-fungal treatment Anti-fungal agents	36 (87.8) Anidulafungin: 16 (44.4) Caspofungin: 13 (36.1) Fluconazole: 12 (33.3) Other azoles: 5 (13.9) AMB: 4 (11.1) Other: 3 (8.3)
Number of deaths	23 (56.1)

Data are absolute numbers (percentages) unless otherwise specified. Abbreviations: ICU, intensive care unit; IQR, interquartile range; TI, tracheal intubation; AMB, amphotericin B.

**Table 5 jof-07-00921-t005:** Observational studies reporting candidemia associated with COVID-19.

Characteristics	No. (%)
No. of candidemia cases/total number of patients	360/9451 (3.8)
Sex	Male: 162 (45.1)
Main underlying diseases	Diabetes mellitus: 91 (25.3) Arterial hypertension: 74 (20.6) Cardiovascular disease: 60 (16.7) Chronic pulmonary disease: 37 (10.3) Obesity: 23 (6.4)
COVID-19 treatments	Steroids: 156 (43.4) Tocilizumab: 69 (19.2)
No. of patients with central venous catheters	247/267 (92.5)
No. of patients on total parenteral nutrition	49/229 (21.4)
No. of patients on antibiotic therapy	172/180 (95.6)
No. of patients undergoing mechanical ventilation	227/271 (83.8)
*Candida* species	*C. albicans:* 164 (44.4) *C. glabrata:* 51 (14) *C. parapsilosis:* 50 (13.8) *C. tropicalis:* 31 (8.5) *C. auris*: 20 (5.5) *C. dubliniensis*: 4 (1.1) Other: 43 (11.8) Total: 363
No. of patients receiving anti-fungal treatment Anti-fungal agents	190/253 (75) Fluconazole: 38 (20) Anidulafungin: 37 (19.4) Caspofungin: 25 (13.2) Micafungin: 20 (10.5) Fluconazole: 38 (20) Other azoles: 5 (2.6)
No. of deaths	202/270 (74.8)

Data are absolute numbers (percentage), unless otherwise specified. Abbreviations: ICU, intensive care unit; IQR, interquartile range; TI, tracheal intubation; AMB, amphotericin B.

**Table 6 jof-07-00921-t006:** Case reports and case series reporting mucomycosis associated with COVID-19.

Characteristics	No. (%)
Mucormycosis cases	59
Sex	Males: 42 (71.2)
Median age (IQR)	51 (42–62)
Main underlying diseases	Diabetes mellitus: 46 (77.9) Arterial hypertension: 14 (23.7) Cardiovascular disease: 6 (10.2) Obesity: 5 (8.4) Chronic pulmonary disease: 2 (3.4)
COVID-19 treatments	Steroids: 33 (55.9) Tocilizumab: 8 (13.6)
Timing of occurrence of mucormycosis	During COVID-19: 50 (84.7) After COVID-19: 9 (15.3)
Mucormycosis site	Rhino-orbital: 19 (32.2) Rhino-orbital-cerebral: 19 (32.2) Pulmonary: 12 (20.3) Rhino-sinusal: 3 (5.1) Disseminated: 2 (3.4) Soft tissues: 1 (1.7) Muscolo-skeletal: 1 (1.7) Gastric: 1 (1.7) Palatal: 1 (1.7)
Histopathological diagnosis, specimen	53/59 (91.5) Nasal biopsy: 29 (54.7) Nasal material: 10 (18.9) Autopsy: 4 (7.5) BAL: 4 (7.5) Lung biopsy: 2 (3.8) Sputum: 1 (1.9) Other biopsies: 5 (9.4)
Positive culture species	33/37 (89.2) *R. arrhizus*: 7 *R. microsporus:* 5 (+1 sample with only positive PCR) *Lichtemia:* 3 *R. azygosporus*: 1 *Rhizopus*/*Mucor* spp.: 13 Species not specified: 4 +1 case with a positive *Mucorales* PCR
No. of patients receiving anti-fungal treatment Anti-fungal agents	55/59 (93.1) AMB: 53 (96.4) Azoles: 20 (36.4) Echinocandins: 3 (5.5)
No. of patients undergoing surgery	35/59 (59.3) Maxillofacial surgery: 31 (88.6) Lung resection: 1 (2.9) Pleural decortication: 1 (2.9) Soft tissue debridement: 1 (2.9)
No. of deaths	30 (50.8)

Data are absolute numbers (percentage) unless otherwise specified. Abbreviations: IQR, interquartile range; AMB, amphotericin B; BAL, bronchial alveolar alavage. Nasal material: nasal samples other than biopsy (i.e., swabs, discharges, etc.).

**Table 7 jof-07-00921-t007:** Observational studies reporting mucormycosis associated with COVID-19.

Characteristics	No. (%)
No. of mucormycosis cases/total number of patients	3126/36509 (8.6)
Sex	Male: 2206 (70.6)
Main underlying diseases	Diabetes mellitus: 2379 (76.1) Arterial hypertension: 731 (23.4) Cardiovascular disease: 38 (1.2) Obesity: 4 (0.1) Chronic pulmonary disease: 27 (0.9)
COVID-19 treatments	Steroids: 2307 (73.8) Tocilizumab: 65 (2.1)
Timing of occurrence	During COVID-19: 231 After COVID-19: 69
Mucormycosis site	Rhino-orbital: 1530 (48.9) Rhino-sinusal: 769 (24.6) Rhino-orbito-cerebral: 635 (20.3) Pulmonary: 20 (0.6) Disseminated: 4 (0.1) Other sites: 4 (0.1) Site not specified: 5 (0.2)
Histopathological diagnosis	1340/3062 (43.8)
Positive culture Species	575/2437 (23.6) *Mucorales* not specified: 100 *Rhizopus* spp.: 11 *R. arrhizus:* 8 *Mucor* spp.: 6 *Lichtemia* spp.: 2 *R. stolonifer*: 1 *R. microsporus*: 1 Species not specified: 508 +1 case with positive *Rhizopus/Mucor* PCR on RS and serum
No. of patients receiving anti-fungal treatment Anti-fungal agents	2352/3113 (75.6) AMB: 2289 (97.3) Posaconazole: 799 (33.9) Isavuconazole: 54 (2.3) Voriconazole: 2 (0.1) Caspofungin: 2 (0.1)
No. of patients undergoing surgery	1808/2646 (68.3)
No. of deaths	405/2524 (16)

Data are absolute numbers (percentage) unless otherwise specified. Abbreviations: AMB, amphotericin B. The two largest studies [176,177] report the median time from the diagnosis of COVID-19 to the diagnosis of COVID-associated mucormycosisbut not the frequency of patients with cuncurrent or post-COVID-19 infection.

**Table 8 jof-07-00921-t008:** Studies reporting rarer invasive fungal infections associated with COVID-19.

Author/ Country/ Study Type	IFI	Cases	Sex/Age	Underlying Diseases	COVID-19 Treatment	Temporal Relationship with COVID-19	Microbiological Data Supporting Diagnosis	Treatment	Outcome
Menon [179]/ USA/ Case report	PCP	1	M/83	CVD, CPD, UC	Steroids IMV	Concurrent	Positive *P. jirovecii* PCR on TA Positive serum BDG	TMP/SMX	Alive
Mang [180]/ Germany/ Case report	PCP	1	M/52	Obesity, AH, AIDS	IMV	Concurrent	Positive *P. jirovecii* PCR on BAL	TMP/SMX	Alive
Bhat [181]/ USA/ Case report	PCP	1	M/25	AIDS	Steroids IMV	Concurrent	Positive *P. jirovecii* DFA on BAL	TMP/SMX	Alive
De Francesco [182]/ Italy/ Case report	PCP	1	M/65	AH, DM, CVD, SOT, chronic steroids	Steroids TCZ IMV	Concurrent	Positive *P. jirovecci* PCR on sputum	TMP/SMX	Dead
Alanio [183]/ France/ Prospective	PCP	10/108 (9.3%)	8 M (80%)/ 59 year (46–68)	Obesity (3), AH (6), DM (3), CVD (1), CPD (1)	Steroids (7) Eculizumab (2) IMV (10)	Concurrent (10)	Positive *P. jirovecii* PCR in RS (10)	TMP/SMX: 4	3 deaths (30%)
Broadhurst [178]/ USA/ Case report	PCP	1	M/54	AH, DM, AIDS	Steroids	Concurrent	Positive *P. jirovecii* DFA in sputum Positive serum BDG	TMP/SMX	Dead
Jeican [184]/ Romania/ Case report	PCP	1	M/52	AH, CVD, liver disease	Steroids IMV	Concurrent (post-mortem diagnosis)	Positive HPE upon autopsy	No	Dead
Coleman [185]/ United Kingdom/ Case report	PCP	1	M/55	CPD, HIV	Steroids	Concurrent	Positive *P. jirovecii* PCR in sputum	TMP/SMX	Alive
Rubiano [186]/ USA/ Case report	PCP	1	M/36	AIDS	IMV	Concurrent	Positive *P. jirovecii* PCR and DFA in BAL Positive serum BDG	TMP/SMX	Dead
Viceconte [187]/ Italy/ Case report	PCP	1	M/50	None	Steroids	Concurrent	*P. jirovecii* DFA on BAL	TMP/SMX	Alive
Larzàbal [188]/ Argentina/ Case report	PCP	1	F/46	AIDS, Raynaud syndrome	Steroids	Concurrent	Positive direct examination of sputum	TMP/SMX	Alive
Total	PCP	20	17 M (85%)	Obesity (4), AH (10), DM (5), CVD (4), CPD (3), HIV (6)	Steroids (15) TCZ (1) Eculizumab (1) IMV (16)	Concurrent (20)	Positive PCR 15: 1 TA, 2 BAL, 2 sputum, 10 RS Positive DFA 4: 3 BAL, 1 sputum Positive serum BDG 3 Histopathology 1 Direct examination 1	TMP/SMX 13	8 deaths (40%)
Khatib [110]/ Qatar/ Case report	Crypto	1	M/60	AH, DM, CVD	Steroids TCZ IMV	Concurrent	*C. neoformans* in blood culture	AMB + flucytosine	Dead
Zhang [137]/ China/ Retrospective	Crypto	1/38 (2.6%)	NR	NR	NR	Concurrent	*Cryptococcus* in blood culture	NR	NR
Woldie [189]/ Canada/ Case report	Crypto	1	M/24	Hemolytic anemia, chronic steroids	Steroids IMV	Concurrent (post-mortem diagnosis)	*C. neoformans* in blood culture	None	Dead
Passarelli [190]/ Brazil/ Case report	Crypto	1	M/75	AH, SOT, chronic steroids	Steroids IMV	Concurrent	*C. neoformans* in blood culture	None	Dead
Ghanem [191]/ USA/ Case report	Crypto	1	F/73	None	Steroids	Concurrent	Positive *Cryptococcal* Ag in cultured CSF	AMB + flucytosine VP derivation	Alive
White [30]/ United Kingdom/ Prospective	Fungemia	1/135 (0.7%)	NR	AH	IMV	Concurrent	*Rhodotorula* in blood culture	AMB Caspofungin	Dead
Fekkar [45]/ France/ Retrospective	Fusariosis	1/260 (0.4%)	M/57	DM, AH	IMV	Concurrent	*F. proliferatum* in BAL culture Positive GM on BAL (1.7)	AMB Caspofungin	Alive
Ashour [148]/ Egypt/ Case report	Fungal sinusitis	1/8 (12.5%)	M/47	DM	IMV	Concurrent	Positive *Aspergillus* culture of nasal biopsy + HPE	AMB Surgery	Alive
Messina [192]/ Argentina/ Case report	Disseminated histoplasmosis	1	F/36	AIDS	NR	Post-COVID	Positive sputum DE for *Histoplasma* Positive serum *Histoplasma* Ag Positive urine *Histoplasma* Ag	AMB Itraconazole	Alive
De Macedo [193]/ Brazil/ Case report	Pulmonary histoplasmosis	1	M/20	No	NR	Post-COVID	Positive serum *Histoplasma* IgM Positive *Histoplasma* PCR on sputum Positive serum *Histoplasma* WB Positive serum GM	Itraconazole	Alive
		1	M/32	No	Steroids	Post-COVID	Positive *Histoplasma* PCR on sputum Positive serum *Histoplasma* WB Positive urine *Histoplasma* Ag Positive serum GM	Itraconazole	Alive
Bertolini [194]/ Argentina/ Case report	Disseminated histoplasmosis	1	M/43	AIDS	NR	Concurrent	Positive HPE of cutaneous lesions *Histoplasma* on blood culture	AMB Itraconazole	Alive
Basso [195]/ Brazil/Case report	Disseminated histoplasmosis	1	F/43	AIDS	Steroids	Concurrent	Positive sputum DE Positive urine *Histoplasma* Ag	Itraconazole	Alive
Shah [196]/ USA/ Case report	CM	1	M/48	Chronic cavitary CM	NR	COVID after CM diagnosis	Positive CM serology	NR	Alive
Rabagliati [61]/Chile/ Observational	*Scedosporium* infection	1	M/89	DM, AH, CPD	None	Cuncurrent	Positive BAL culture	Isavuconazole	Alive

Abbreviations: AH, arterial hypertensione; CVD, cardiovascular disease; CPD, chronic pulmonary disease; UC; ulcerative colitis; AIDS, acquired immuno-deficiency syndrome; IMV, invasive mechanical ventilation; TA, tracheal aspirate; BAL, bronchial alveolar lavage; PCP, P. jirovecii pneumoniae; BDG, (1,3)-beta-D-glucan; TMP/SMX, trimethoprim/sulfamothoxazole; SOT; solid organ transplantation; DM, diabetes mellitus; CM, coccidioidiomycosis; Ag, antigen; WB, Western blot, GM, galactomannan; HPE, histopathological examination; VP, ventriculo-peritoneal, DFA, direct fluorescent antigen.

## Data Availability

No new data were created or analysed in this study. Data sharing does not apply to this article.

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
