# Peer review of "Invasive Fungal Infections Complicating COVID-19: A Narrative Review"

_jof, 2021, doi:10.3390/jof7110921_

Round 1

Reviewer 1 Report

The authors have collected the available data on invasive fungal infections complicating COVID-19 and present case reports/case series and observational studies for each infection. This work is considerable given the high number of published data. It is not evident that case reports/case series are useful to list for some infections when there is a lot of observational studies.

Inclusion of fungal infections other than CAPA appears original versus the different already published systematic reviews on CAPA. However, this orientation induces complexity and multiplication of messages. It is a narrative descriptive review. Research strategy is well presented, a lot of recent papers are cited. All the tables are descriptive and, consequently, long. Synthetic tables would be useful.

  1. For the CAPA part

CAPA is a particular infection related to the different case definitions.

Tables 1 and 2: direct examination was not presented among the retained criteria (columns), however, it may be important to define the CAPA. At least, if this information is available, state the direct examination result combined with the culture result.

The major point is the table 3 with the presentation of the different definitions on CAPA. It has to be specified and improved.

First, it would be interesting to specify the date of each reference on aspergillosis definitions. Consequently, Schauwvliege criteria may be presented before AspICU Verweij’s criteria. The both criteria were defined for influenza, not for Covid-19, then it should be specified that their algorithms were applied to another virus infection.

For modified AspICU Verweij’s criteria, in the comments: “no host risk factors among the diagnostic criteria” it is rather lack of traditional EORTC host factors but there were pre-existing lung disease and immunosuppressive treatment.

For the modified AspICU IAPA criteria of Schauwvliege, this author did not define probable invasive aspergillosis but putative invasive aspergillosis.

For the ECMM/ISHAM definitions:

- add “NA” for putative invasive aspergillosis line.

- For probable and possible aspergillosis definitions, CT scan would be better in an “Imaging or radiological” part rather than “clinical factors”, as used for the other definitions (even the definition authors used “clinical factors”). The retained clinical factors have to be given.

- Nodules are also included in the imaging criteria.

- For probable invasive aspergillosis, mycological evidence was at the end: or “non-BAL GM index > 1.2” (and not > 1) plus another…

- Culture is also lacking among the other criterium associated to “non-BAL GM index > 1.2”.

For the EORTC/MSG criteria (Donnelly): proven IA with histopathologic, cytopathologic or direct microscopic examination of “specimen obtained by a sterile procedure from a normally sterile site” (Idem for the culture). For the PCR, DNA sequencing was performed “on fixed tissues”.

Among the mycological evidence for probable IA, there is also “microscopical detection of fungal elements in sputum, BAL, bronchial brush, or aspirate indicating a mold”.

Since the authors classify the patients from the different CAPA studies in table 4 using the table 3, this table 4 has to be checked.

  1. For the other parts, it would be interesting to specify the geographical origin of the studies in presentation paragraphs, as performed for the CAPA part. These elements are also to discuss for all infections (lacking for CAPA). These parts are easier to follow since case definitions appeared less complex than for CAPA.

In the discussion the authors cited reference 202 considering only observational studies which used rigorous diagnostic criteria, would the authors explain which rigorous criteria were used compared to the table 3 CAPA definitions? Moreover, the authors have to discuss about the direct examination of respiratory samples.

Could you correct “positive Pneumocystis PCR on oropharyngeal wash samples (most common diagnostic sample used) probably overestimate the real burden of the disease”? Oropharyngeal wash samples are not the most common sample for Pneumocystis infection diagnosis, compared to expectoration or BAL.

In the last sentence, do the authors think that “an extra effort to obtain histopathological confirmation to reach a definite diagnosis” would be possible respecting the ethical conditions?

Minor comments:

  • In the part “case report and case series”, it is better to separate BAL from the other respiratory samples (as it is performed for the observational studies) since higher respiratory samples are not really validated for galactomannan detection. Cut off for these last samples were not defined and are probable higher than in BAL then the medians have to be calculated separately.
  • Moreover, the authors indicate that an index > 1 was considered as positive for respiratory samples. What are their reference?
  • Table 1/corresponding text:

In the text, serum GM was positive (ODI>0.5) in 26/49 cases, however, in the table 1, there were only 47 cases with GM ODI and index were available in 21 cases. Moreover, GM was reported in the text positive (ODI>1) in respiratory samples in 26/28 cases. In the table, there were only 23 cases with respiratory GM ODI>1. Could you explain that, please?

  • The authors have interest to discuss with a mycologist about the fungal names:
  • “Hystoplasm*” choosen as research term in Pudmed (page 3), instead of “histoplasm*”
  • Trichosporon asahii, not Thricosporon
  • Rhizopus oryzae not oryzhae or oryzahe… Current name now is Rhizopus arrhizus
  • Rhizopus azygosporus not azygospouros
  • Raoultella mucillaginosa does not exist (Raoultella are bacteria), it is probably Rhodotorula mucilaginosa
  • Lichtheimia, not Absidia

Table 7, reference 166, for this patient, the original authors indicate a positive culture with Lichtheimia (Absidia is an ancient term). Please, correct it in the reference line and in the total.

Table 8, reference 61, three cases of mucormycosis with three fungal culture, one of them being identified as Scedosporium (in pure culture cf reference 61, appendix data). Are the authors considering Scedoporium as a mucormycosis agent?

Reference 202 Title starts by “Incidence”, not “The incidence”

Reference 211 is given without the journal’s name

Other corrections to perform: “fifithy”, “rethinitis”, “aimint”

Author Response

Reviewer #1

The authors have collected the available data on invasive fungal infections complicating COVID-19 and present case reports/case series and observational studies for each infection. This work is considerable given the high number of published data. It is not evident that case reports/case series are useful to list for some infections when there is a lot of observational studies.

The aim of our review is to describe clinical characteristic of IFIs complicating COVID-19 clinical course. Even if observational studies report a higher number of cases, the detailed information given in case report and case series may be useful to clinicians in managing such patients.

Inclusion of fungal infections other than CAPA appears original versus the different already published systematic reviews on CAPA. However, this orientation induces complexity and multiplication of messages. It is a narrative descriptive review. Research strategy is well presented, a lot of recent papers are cited. All the tables are descriptive and, consequently, long. Synthetic tables would be useful.

We are grateful to the referee for this comment. We include synthetic tables in the main article. The longest tables with detailed information about single studies are now included in Supplementary material.

For the CAPA part

CAPA is a particular infection related to the different case definitions.

Tables 1 and 2: direct examination was not presented among the retained criteria (columns), however, it may be important to define the CAPA. At least, if this information is available, state the direct examination result combined with the culture result.

Data about direct examination of respiratory specimens were reported in Table 1-2-S1-S2 and in the main text.

The major point is the table 3 with the presentation of the different definitions on CAPA. It has to be specified and improved.

First, it would be interesting to specify the date of each reference on aspergillosis definitions. Consequently, Schauwvliege criteria may be presented before AspICU Verweij’s criteria.

The both criteria were defined for influenza, not for Covid-19, then it should be specified that their algorithms were applied to another virus infection.

For modified AspICU Verweij’s criteria, in the comments: “no host risk factors among the diagnostic criteria” it is rather lack of traditional EORTC host factors but there were pre-existing lung disease and immunosuppressive treatment.

For the modified AspICU IAPA criteria of Schauwvliege, this author did not define probable invasive aspergillosis but putative invasive aspergillosis.

Table 3 (now Table S3 in the current manuscript version) was revised according to referee’s suggestions.

For the ECMM/ISHAM definitions:

- add “NA” for putative invasive aspergillosis line.

- For probable and possible aspergillosis definitions, CT scan would be better in an “Imaging or radiological” part rather than “clinical factors”, as used for the other definitions (even the definition authors used “clinical factors”). The retained clinical factors have to be given.

- Nodules are also included in the imaging criteria.

- For probable invasive aspergillosis, mycological evidence was at the end: or “non-BAL GM index > 1.2” (and not > 1) plus another…

- Culture is also lacking among the other criterium associated to “non-BAL GM index > 1.2”.

For the EORTC/MSG criteria (Donnelly): proven IA with histopathologic, cytopathologic or direct microscopic examination of “specimen obtained by a sterile procedure from a normally sterile site” (Idem for the culture). For the PCR, DNA sequencing was performed “on fixed tissues”.

Among the mycological evidence for probable IA, there is also “microscopical detection of fungal elements in sputum, BAL, bronchial brush, or aspirate indicating a mold”.

ECMM/ISHAM definitions are now correctly reported as originally stated by the authors of the paper.

Since the authors classify the patients from the different CAPA studies in table 4 using the table 3, this table 4 has to be checked.

Table 4 (Table 3 in the current manuscript version) was checked, and modifications are tracked in the main text.

For the other parts, it would be interesting to specify the geographical origin of the studies in presentation paragraphs, as performed for the CAPA part. These elements are also to discuss for all infections (lacking for CAPA). These parts are easier to follow since case definitions appeared less complex than for CAPA.

The geographical origin of the studies is now reported for all IFI type.

In the discussion the authors cited reference 202 considering only observational studies which used rigorous diagnostic criteria, would the authors explain which rigorous criteria were used compared to the table 3 CAPA definitions?

The term “rigorous” was indeed misleading. Now we used the term “well-defined” diagnostic criteria. Ref. 202 considers studies in which validated diagnostic algorithms (listed in Table 3) are used. This is in contrast with some case report or case series, where the diagnostic algorithm is not always reported or where authors use personalized clinical algorithms to diagnose CAPA.

Moreover, the authors have to discuss about the direct examination of respiratory samples.

Discussion about direct examination of respiratory samples was included in section 4.

Could you correct “positive Pneumocystis PCR on oropharyngeal wash samples (most common diagnostic sample used) probably overestimate the real burden of the disease”? Oropharyngeal wash samples are not the most common sample for Pneumocystis infection diagnosis, compared to expectoration or BAL.

The main text was corrected according to referee comment.

In the last sentence, do the authors think that “an extra effort to obtain histopathological confirmation to reach a definite diagnosis” would be possible respecting the ethical conditions?

The diagnosis of CAPA, mainly in ICU patients, is difficult and it has often a considerable degree of uncertainty, given the diagnostic tools we have. Considering the severity of critically ill COVID-19 patients, it is not uncommon that clinical suspicion of CAPA leads to initiate antifungal therapy. Antifungal therapy is protracted for weeks, and it is difficult to evaluate the clinical response to antifungals. Furthermore, such drugs can cause many adverse events. Thus, we think that obtaining a definite diagnosis would be essential to treat the patients that really need a specific antifungal therapy.

Minor comments:

In the part “case report and case series”, it is better to separate BAL from the other respiratory samples (as it is performed for the observational studies) since higher respiratory samples are not really validated for galactomannan detection. Cut off for these last samples were not defined and are probable higher than in BAL then the medians have to be calculated separately.

Manuscript was modified according to referee comments.

Moreover, the authors indicate that an index > 1 was considered as positive for respiratory samples. What are their reference?

Cut-off of galattomannan positivity are stated by ESCMID guidelines about diagnosis and management of Aspergillus diseases. Ulmann, et al. Diagnosis and management of Aspergillus diseases: executive summary of the 2017 ESCMID-ECMM-ERS guideline. Clin Microbiol Infect. 2018;24:1. https://doi.org/10.1016/j.cmi.2018.01.002.

Table 1/corresponding text:

In the text, serum GM was positive (ODI>0.5) in 26/49 cases, however, in the table 1, there were only 47 cases with GM ODI and index were available in 21 cases. Moreover, GM was reported in the text positive (ODI>1) in respiratory samples in 26/28 cases. In the table, there were only 23 cases with respiratory GM ODI>1. Could you explain that, please?

Positivity rate of GM in serum and respiratory samples was wrong. Correct data are now reported in Table 1-2 and in the main text. Positivity rate was calculated irrespectively of ODI discrete value availability (some papers report a positive GM, i.e. above the cut-off, but without the precise value). Median ODI was then calculated including only the studies with a GM precise value.

The authors have interest to discuss with a mycologist about the fungal names:

“Hystoplasm*” choosen as research term in Pudmed (page 3), instead of “histoplasm*”

This was a spelling mistake in the manuscript, which is now correct.

Trichosporon asahii, not Thricosporon

This was a spelling mistake in the manuscript, which is now correct.

Rhizopus oryzae not oryzhae or oryzahe… Current name now is Rhizopus arrhizus

Rhizopus azygosporus not azygospouros

Lichtheimia, not Absidia

Table 7, reference 166, for this patient, the original authors indicate a positive culture with Lichtheimia (Absidia is an ancient term). Please, correct it in the reference line and in the total.

In the first version of the manuscript, species identification was reporting according to original author terminology. We agree with the referee to change the species names according to the most recent taxonomy. The tables and the main text are now updated.

Raoultella mucillaginosa does not exist (Raoultella are bacteria), it is probably Rhodotorula mucilaginosa.

The reviewer is correct and we apologize for the mistake that has been  corrected.

Table 8, reference 61, three cases of mucormycosis with three fungal culture, one of them being identified as Scedosporium (in pure culture cf reference 61, appendix data). Are the authors considering Scedoporium as a mucormycosis agent?

Scedosporium infection was initially reported among mucormycosis because it was included in one of the observational studies about CAM. We agree with the referee that Scedosporium infection is distinct from mucormycosis. It is now stated in Table 8 (rare infections). Text was also modified.

Reference 202 Title starts by “Incidence”, not “The incidence”

Reference 211 is given without the journal’s name

Other corrections to perform: “fifithy”, “rethinitis”, “aimint” 

The text was modified according to referee suggestions.

Reviewer 2 Report

This is a really good, useful and quite detailed article.

Major comments:

More data should be reported about SARS-CoV-2, since this is the main condition leading to the fungal infections that are further analysed. Any correlation between SARS-CoV-2 (variant, viral load, time of infection etc) and the frequency or the severity of fungal complications should be reported.

Minor comments:

1. Attention to spelling. Examples: "an high" → "a high" (Abstract), "indipendent risck" → "independent risk" (Discussion).

2. Table 1. The definition of "ALL" should be added to Abbreviations.

Author Response

Reviewer #2

This is a really good, useful and quite detailed article.

We thank the reviewer for her/his appreciation of our work

Major comments:

More data should be reported about SARS-CoV-2, since this is the main condition leading to the fungal infections that are further analysed. Any correlation between SARS-CoV-2 (variant, viral load, time of infection etc) and the frequency or the severity of fungal complications should be reported.

Frequency and severity of fungal infections are reported in the text. Virological data about SARS-CoV-2 infection(i.e. variant, viral load, etc.)  are rarely mentioned in the studies, so we decided not to report them. The focus of this review is to describe IFI clinical characteristics. 

Minor comments:

  1. Attention to spelling. Examples: "an high" → "a high" (Abstract), "indipendent risck" → "independent risk" (Discussion).
  2. Table 1. The definition of "ALL" should be added to Abbreviations.

The text was modified according to referee suggestion.

Reviewer 3 Report

Thank you for the opportunity to review your manuscript intitled “Invasive fungal infections (IFIs) complicating COVID-19: a narrative review”.

In this paper, the authors reviewed case report, case series, and cohort studies about aspergillosis, candidemia, mucormycosis, pneumocystosis, and rares IFIs in COVID-19 patients, published between 1 January 2020 and 16 June 2021.

The main findings are:

  • IFIs are secondary infections that can complicate critical COVID-19 and are associated with increased mortality.
  • CAPA has an average incidence of 3.3%, is mainly related to fumigatus, and is associated with diagnosis and management issues. Risk factors and clinical presentation are different from aspergillosis usually seen in intensive care patients. Medical history of CAPA patients mainly included cardiopulmonary chronic diseases (34.9%), systemic corticosteroids (34.4%) and diabetes mellitus (28.6%). CAPA classification criteria are not the same among the various existing guidelines.
  • The average incidence of candidemia is 3.8% (mainly albicans). Classic risk factors were present in almost all patients with CAC.
  • The average incidence of mucormycosis is 0.9%. CAM is associated with poorly controlled diabetes mellitus and steroids use. Rhino-orbital and rhino-orbitalcerebral were the most common localizations.
  • Pneumocystis jirovecii pneumonia, cryptococcal infections and fusariosis are rare IFIs associated with COVID-19.

This paper is interesting, findings are well described and the manuscript is appropriately organized.

I have the following comments:

  • My main concern is about the novelty of the present paper. Indeed, numerous reviews have been published on this topic in 2021…
  • Please remove abbreviation "IFIs" from the title 
  • The authors should insist more on the fact that the IFIs overwhelmingly occur in ICU patients, e. in those with critical COVID-19. Especially, this fact is not emphasized enough in the abstract, nor in the conclusion.
  • Tables 3 is informative but too large and therefore difficult to read. Another presentation of this table is mandatory. Table 4 is even larger and difficult to read and it is less interesting... Its place could be in the appendix.
  • The largest study about COVID-19 associated mucormycosis (CAM), including more than 2800 patients has been published only 2 days after the end of the research period of the present study (Sen et al . Epidemiology, clinical profile, management, and outcome of COVID-19-associated rhino-orbital-cerebral mucormycosis in 2826 patients in India – Collaborative OPAI-IJO Study on Mucormycosis in COVID-19 (COSMIC), Report 1. Indian J Ophthalmol 2021;69:1670-92.). This study was not included in the authors' work, which limits the weight of the conclusions of this paper on CAM... Really shame for only 2 days !
  • CAM is an emerging health concern in countries with a high background prevalence of mucormycosis (India, Iran…) with diagnosis and management issues similar to CAPA. A comparison of existing guidelines, as the authors did for CAPA, might be useful (Rudramurthy et al; ECMM/ISHAM recommendations for clinical management of COVID-19 associated mucormycosis in low- and middle-income countries. Mycoses. 2021;64:1028–1037. https://doi.org/10.1111/myc.13335 ; Malhotra et al; COVID-19 associated mucormycosis: Staging and management recommendations (Report of a multi-disciplinary expert committee). Journal of Oral Biology and Craniofacial Research 11 (2021) 569–580. https://doi.org/10.1016/j.jobcr.2021.08.001)

Thank you again for the opportunity to read this work.

Author Response

Reviewer #3

Thank you for the opportunity to review your manuscript intitled “Invasive fungal infections (IFIs) complicating COVID-19: a narrative review”.

In this paper, the authors reviewed case report, case series, and cohort studies about aspergillosis, candidemia, mucormycosis, pneumocystosis, and rares IFIs in COVID-19 patients, published between 1 January 2020 and 16 June 2021.

The main findings are:

IFIs are secondary infections that can complicate critical COVID-19 and are associated with increased mortality.

CAPA has an average incidence of 3.3%, is mainly related to fumigatus, and is associated with diagnosis and management issues. Risk factors and clinical presentation are different from aspergillosis usually seen in intensive care patients. Medical history of CAPA patients mainly included cardiopulmonary chronic diseases (34.9%), systemic corticosteroids (34.4%) and diabetes mellitus (28.6%). CAPA classification criteria are not the same among the various existing guidelines.

The average incidence of candidemia is 3.8% (mainly albicans). Classic risk factors were present in almost all patients with CAC.

The average incidence of mucormycosis is 0.9%. CAM is associated with poorly controlled diabetes mellitus and steroids use. Rhino-orbital and rhino-orbitalcerebral were the most common localizations.

Pneumocystis jirovecii pneumonia, cryptococcal infections and fusariosis are rare IFIs associated with COVID-19.

This paper is interesting, findings are well described and the manuscript is appropriately organized.

We thank the reviewer for her/his appreciation of our work

I have the following comments:

My main concern is about the novelty of the present paper. Indeed, numerous reviews have been published on this topic in 2021…

We know that IFI in COVID-19 is a trending topic nowadays. However, we think that our study is unique because is the only descriptive review in which all types of IFIs associated with COVID-19 are considered and analysed. Furthermore, the number of patients and cases reported is high.

Please remove abbreviation "IFIs" from the title

The text was modified according to referee suggestions.

The authors should insist more on the fact that the IFIs overwhelmingly occur in ICU patients, e. in those with critical COVID-19. Especially, this fact is not emphasized enough in the abstract, nor in the conclusion.

A paragraph about the burden of IFI in ICU patients was added to abstract and conclusions.

Tables 3 is informative but too large and therefore difficult to read. Another presentation of this table is mandatory. Table 4 is even larger and difficult to read, and it is less interesting... Its place could be in the appendix.

We agree with the reviewer’s comments, the longest Tables are now shifted in Supplementary materials.

The largest study about COVID-19 associated mucormycosis (CAM), including more than 2800 patients has been published only 2 days after the end of the research period of the present study (Sen et al . Epidemiology, clinical profile, management, and outcome of COVID-19-associated rhino-orbital-cerebral mucormycosis in 2826 patients in India – Collaborative OPAI-IJO Study on Mucormycosis in COVID-19 (COSMIC), Report 1. Indian J Ophthalmol 2021;69:1670-92.). This study was not included in the authors' work, which limits the weight of the conclusions of this paper on CAM... Really shame for only 2 days !

We have changed the period of search (extended to 18 June 2021) to include the study indicated by the reviewerThis large study is now included in our review. The incidence rate and characteristic of CAM patients were recalculated.

CAM is an emerging health concern in countries with a high background prevalence of mucormycosis (India, Iran…) with diagnosis and management issues similar to CAPA. A comparison of existing guidelines, as the authors did for CAPA, might be useful (Rudramurthy et al; ECMM/ISHAM recommendations for clinical management of COVID-19 associated mucormycosis in low- and middle-income countries. Mycoses. 2021;64:1028–1037. https://doi.org/10.1111/myc.13335 ; Malhotra et al; COVID-19 associated mucormycosis: Staging and management recommendations (Report of a multi-disciplinary expert committee). Journal of Oral Biology and Craniofacial Research 11 (2021) 569–580. https://doi.org/10.1016/j.jobcr.2021.08.001)

Following referee suggestions, we expanded the discussion of mucormycosis by including relevant information about diagnostic and treatment culprit. The two articles suggested are now in the reference list.

Thank you again for the opportunity to read this work.

Reviewer 4 Report

The authors in the manuscript provide a comprehensive review of the presence of invasive fungal infections in patients with COVID-19 disease, The authors describe their search methods  for retriveing articles, The report data at the patient level, article level and provide detailed information about the studies.  In their discussion they highlight some of the challenges encountered in the various IFI in COVId-19. They also present the limitation of diagnosis of different IFI by currently proposed lmethod for CAPA.

The mauscript is very long and difficult to read alot of this is from the tables where alot of detal has been captured some table example Table 1 can be added as a supplemetary materia and the information boldened at the bottom of the table which summarizes the table be used in the text.

There are some  spelling or punctuation errors examples include

  • In abstract there were two periods after the word very rare and before Candidemia in the 8th line.
  • In introduction the word of is used insteasd of for
  • authors always appears capitalized wherevere it appears in the manuscript

In other areas repharasing of the sentences nay make meaning clearer

example " Because of increased reports of IPA in ICU patients without immunological disorders (i.e., severe influenza and chronic obstructive pulmonary disease) [7,8], since the emergence of COVID-19 in Wuhan, some Authors have highlighted the risk of invasive fungal infections (IFIs) in critically ill COVID-19 patients [9]."

or "Finally, Pneumocystis jirovecii pneumonia (PCP), cryptococcosis and histoplasmosis are other IFIs which have been reported in COVID-19 patients, suggesting that SARSCoV-2 induced immune dysregulation can be as deep to increase the likelihood to develop opportunistic infections which are typically seen in patients with severe immunodepression (i.e., acquired immunodeficiency syndrome [AIDS] or haematological 
malignancies) [19]." the author may consider removinr the word and rephrase statemet,

While a comprehensive review of the literature has been done as a result of the lengthy nature of the paper and the difficulties of the diagnosis of IFI in COVID-19 which was addressed in paper is not really emphasized and other important aspects that clinicians nust bear in mind may be lost to the audience in reading the paper. The paper will benefit from shortening it and point to bring out the limitations in the current dignostic methods for CAPA and other IFIs in COVID-19 (which was doen) highlighted.

Author Response

Reviewer #4

The authors in the manuscript provide a comprehensive review of the presence of invasive fungal infections in patients with COVID-19 disease, The authors describe their search methods  for retriveing articles, The report data at the patient level, article level and provide detailed information about the studies.  In their discussion they highlight some of the challenges encountered in the various IFI in COVId-19. They also present the limitation of diagnosis of different IFI by currently proposed method for CAPA.

The manuscript is very long and difficult to read alot of this is from the tables where alot of detal has been captured some table example Table 1 can be added as a supplemetary materia and the information boldened at the bottom of the table which summarizes the table be used in the text.

We agree with the reviewer’s comment about the readability of the manuscript. According to referees’ comments, the longest Tables are now shifted in Supplementary materials.

There are some  spelling or punctuation errors examples include

In abstract there were two periods after the word very rare and before Candidemia in the 8th line.

In introduction the word of is used insteasd of for ?!

authors always appears capitalized wherevere it appears in the manuscript

In other areas repharasing of the sentences nay make meaning clearer

example " Because of increased reports of IPA in ICU patients without immunological disorders (i.e., severe influenza and chronic obstructive pulmonary disease) [7,8], since the emergence of COVID-19 in Wuhan, some Authors have highlighted the risk of invasive fungal infections (IFIs) in critically ill COVID-19 patients [9]."

or "Finally, Pneumocystis jirovecii pneumonia (PCP), cryptococcosis and histoplasmosis are other IFIs which have been reported in COVID-19 patients, suggesting that SARSCoV-2 induced immune dysregulation can be as deep to increase the likelihood to develop opportunistic infections which are typically seen in patients with severe immunodepression (i.e., acquired immunodeficiency syndrome [AIDS] or haematological

malignancies) [19]." the author may consider removing the word and rephrase statement

Spelling and punctuation mistakes were corrected according to referees’ suggestions.

While a comprehensive review of the literature has been done as a result of the lengthy nature of the paper and the difficulties of the diagnosis of IFI in COVID-19 which was addressed in paper is not really emphasized and other important aspects that clinicians nust bear in mind may be lost to the audience in reading the paper. The paper will benefit from shortening it and point to bring out the limitations in the current dignostic methods for CAPA and other IFIs in COVID-19 (which was doen) highlighted.

The structure of the manuscript was revised according to referees’ suggestion. The longest Tables are now in supplementary material; thus, the paper is now shorted. Synthetic tables about each IFI are now reported and should help the reader to find relevant information.

Round 2

Reviewer 4 Report

No further comments

Author Response

The article needs to be edited by a native English speaker: there are grammatical errors throughout the paper that need to be corrected before publication.

The article has been extensively revised by a native English speaker (also supplementary tables)

I would recommend deleting the reference to mucor as so-called “black fungus”, which is a misnomer.

This misnomer has been appropriately changed